# The hot hand in the wild

**Konstantinos Pelechrinis**[1]*, **Wayne Winston**[2]

**1** Department of Informatics and Networked Systems, University of Pittsburgh, Pittsburgh, PA, United States of America, **2** Kelley School of Business, Indiana University, Bloomington, IN, United States of America

* kpele@pitt.edu

## Abstract

Streaks of success have always fascinated people and a lot of research has been conducted to identify whether the "hot hand" effect is real. While sports have provided an appropriate platform for studying this phenomenon, the majority of existing literature examines scenarios in a vacuum with results that might or might not be applicable *in the wild*. In this study, we build on the existing literature and develop an appropriate framework to quantify the extent to which success can come in streaks—beyond the stroke of chance—in a natural environment. Considering in-game basketball game situations, our analysis provides statistical evidence that individual players do indeed exhibit the hot hand in varying degrees, that is, individual players can consistently get in a streak of successful shots beyond random chance. However, as a whole, the average player exhibits shooting regression, that is, after consecutive makes he tends to perform below expectations. Even though our results are based on a sports setting, we believe that our study provides a path towards thinking of the hot hand beyond a laboratory-like, controlled environment. This is crucial if we want to use similar results to enhance our decision making and better understand short and long term outcomes of repeated decisions.

## Introduction

Recently a video showing Steph Curry, the Golden State Warriors point guard, making 105 consecutive three-pointers during a practise session went viral [1]. Soon after people were trying to estimate the probability of this happening, and they came to the conclusion that in order for this to even have a probability greater than 50% of happening, Curry's true shooting percentage on that specific shot should be 99.5% [2]! This video also triggered again the discussion about the existence or not of the "hot hand", with analysts using this video as evidence that the hot hand truly exists [3, 4].

From the classic work of Gillovich, Vallone and Tversky (GVT) [5], to the most recent major development from Miller and Sanjurjo [6] that reversed the GVT conclusions, the hot hand has occupied researchers in social, statistical and computational sciences. Sports have had a prominent position in these studies since they provide a native environment for studying the streaks of success. Even though the vast majority of studies have focused on basketball (the original setting of the GVT study), other sports have seen their share of hot hand studies [7–10]. Recently similar studies have been conducted for understanding streaks in creative and scientific careers

**Data Availability Statement:** All data and code for their analysis is publicly available at: https://github.com/kpelechrinis/HotHand-NBA-Tracking.

**Funding:** APC charges for this article were fully paid by the University Library System, University of Pittsburgh.

**Competing interests:** The authors have declared that no competing interests exist.

[11]. Dorsey-Palmateer and Smith [7] point out an important, data-related, shortcoming in the original GVT study—and many that came after—in the treatment of the different shots as identical. GVT themselves mentioned possible confounding factors (e.g., taking tougher shots, defensive adjustments etc.) when analyzing shot data from games. They overcame this problem by using free throw data—that can be thought of as identical attempts—and controlled shooting field experiments. This is in general the approach taken by the majority of the literature when studying the phenomenon in basketball, i.e., controlled shooting experiments or analysis of streaks at free throw attempts or three-point contests. These settings are certainly not theoretical and are based on actual performance of players—professional or amateurs—allowing us to observe in practise what one might have expected/estimated in theory. Nevertheless, it should be evident that these settings are not representative of *in-game situations, where different shots are taken under different conditions and hence, have different probability of success*. In fact, the aforementioned practise session video with Curry's 105 consecutive made three-point shots can also be thought of as a controlled environment, given that all shots are taken under the same conditions (ignoring the potential fatigue during his 100th shot, as compared to say his first one). Hence, it is not clear whether the conclusions from studies based on a controlled environment can be generalized to situations experienced *in-game*. In this study, we will eliminate this assumption by utilizing a dataset from shots taken during NBA games. There is very little work dealing with the *truth* or fallacy of the hot hand during in-game situations. In the closest setting to ours, Lantis and Nesson [12] used regression specifications to model the outcome of the next shot, and found that overall there is a negative correlation between a streak of made shots and a make for the next shoot. This is in agreement with our league-wide results, which were obtained with a different analytical framework (we discuss further the work by Lantis and Nesson [12] in the Discussion section).

The rich metadata included will allow us to account for the differences in the shot quality between consecutive shots. Quantifying the quality of a shot (or the shooting ability of a player) has drawn attention in the era of tracking data. Second Spectrum—the NBA's provider of tracking data—has developed its own proprietary model (quantified Shot Quality—qSQ) [13]. Franks *et al.* [14] and Cervone *et al.* [15] used information from these tracking data to build a hierarchical logistic regression model for the success probability of a shot. Similar hierarchical models have been shown to be able to discriminate better the shooting ability of players by shrinking empirical estimates toward more reasonable prior means [16]. While these type of models make use of information at the time of shot release, more recent models use the full trajectory of the ball post-release [17, 18]. These models improve the accuracy of the models, at the cost of requiring ball trajectory information. However, estimating the quality of a shot, i.e., the shot make probability, introduces another challenge in the analysis. In particular, the shot make probability model will inevitably introduce an error in the estimation. Hence, we also introduce a mechanism in our framework that accounts and adjusts for this error.

For our analysis, we will build and expand on the seminal work by Miller and Sanjurjo [6]. They showed that the GVT study—and others following it—suffer from the streak selection bias (explained in detail in the Methods section) that leads to understimation of the hot hand phenomenon. Combined with the aforementioned non-identical property of the trials in our setting, existing approaches can lead to biased estimates of the hot hand effect from in-game situations. We develop a framework (details provided in the Methods) that can be applied to situations appearing *in the wild*. In particular, we extend the empirical test introduced by Miller and Sanjurjo [6] to cases where the individual trials of a sequence are not identically distributed. While we apply our test on data from basketball games, it should become evident that it is applicable on any binary sequence obtained through non-identical trials.

## Materials and methods

We will start by providing the intuition behind the important observation from Miller and Sanjurjo [6]. They pointed out that the GVT study—and many others with a similar analytical approach—suffered from the streak selection bias that while it appears in all sample sizes, is particularly pronounced in small samples. Based on the streak selection bias, if we take a finite series of makes (M) and misses (X) for a sequence of independent trials with constant success probability of 0.5, and we randomly choose one of the makes, then the probability that the next attempt is also a make is lower than 0.5. For example, if we consider a sequence of 20 shots with exactly 10 makes and 10 misses, then once we randomly select one of the makes, the probability that the next shot is a make is $\frac{9}{19} \approx 0.47$. According to Miller and Sanjurjo [6] a good way to answer the debate of hot hand is to perform a permutation test on the sequence of makes and misses and compare the observed probability $\Pr[M|\underbrace{M \ldots M}_{k \; times}]_{data}$ with the empirical distribution of $\Pr[M|\underbrace{M \ldots M}_{k \; times}]_{perm}$ obtained through a number of permutations (we used 500 permutations in our results). This will allow one to test the null hypothesis of no hot hand, i.e., $H_0 : \Pr[M|\underbrace{M \ldots M}_{k \; times}]_{data} = \Pr[M|\underbrace{M \ldots M}_{k \; times}]_{perm}$. This null essentially states that the probability of observing a make following $k$ consecutive makes in the data is the same as the one expected if we permuted the sequence randomly, and hence, there is no serial correlation between makes. For example, let us consider the sequence of shots shown in Fig 1. Let us further assume that all shots are from the same spot (e.g., practise right corner 3 shots such as the ones from Steph

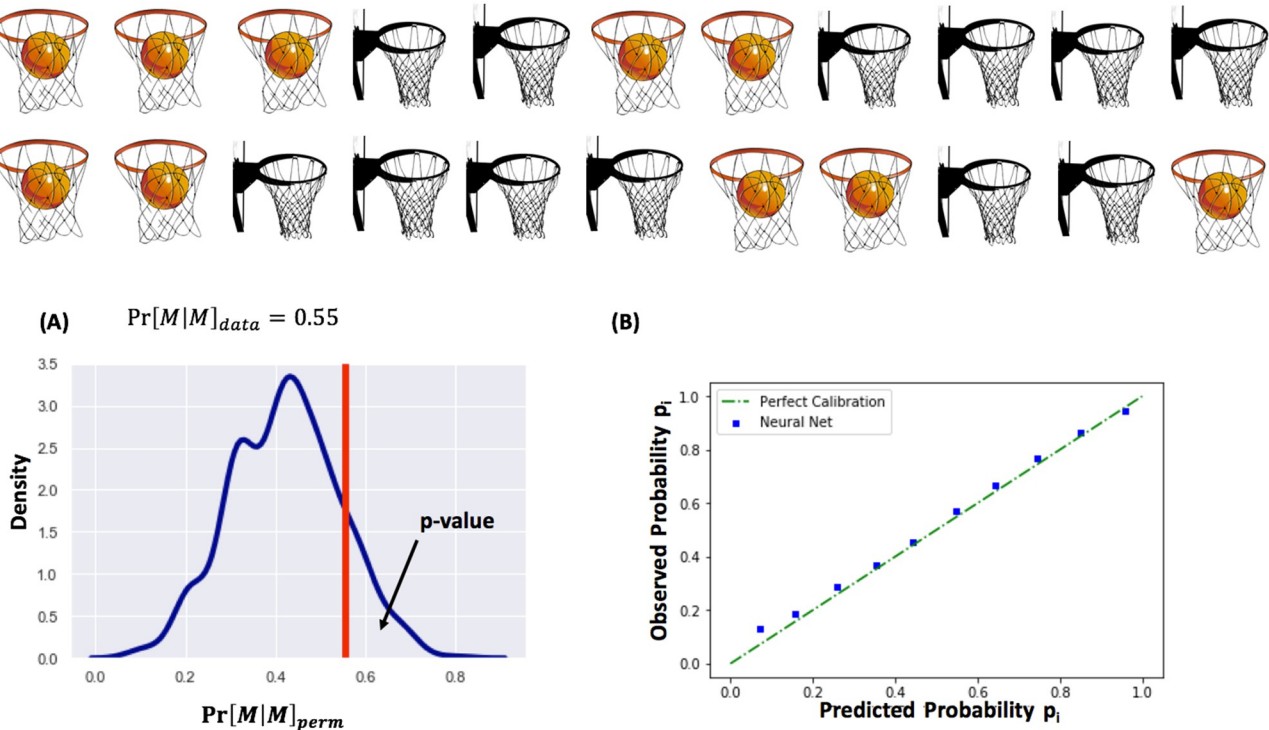

**Fig 1. (A)** The permutation test [6] removes the streak selection bias in small sequences. **(B)** In an actual game environment we need to consider the shot quality. Our shot make probability model outputs accurate out-of-sample probabilities.

Curry). For $k = 1$, we we have $\Pr[M|M]_{data}$ to be equal to 0.55. We randomly permuted the shots 500 times and we obtained the distribution for $\Pr[M|M]_{perm}$ presented at inset (**A**). The average of $\Pr[M|M]_{perm}$ is 0.42—which, is smaller than the base success rate in the sequence of $\frac{10}{22} = 0.45$. Furthermore, in only 5% of the permutations we observed $\Pr[M|M]_{perm} > \Pr[M|M]_{data}$. This is essentially the empirical p-value of the permutation test, and one can say that there is statistical evidence that this sequence exhibits the hot hand. If we use the Wald-Wolfowitz test [19] on the same sequence, which is what the GVT study used, starting with the assumption that the null hypothesis is true—i.e., there is no hot hand—we obtain a p-value of 0.1. This essentially translates to a probability of the observed sequence being a result of chance that is twice as large as compared to the permutation test, thus underestimating the phenomenon.

We would like to emphasize here that our objective for presenting this example is to provide some insight on the sources of streak selection bias, which can be non-intuitive and surprising at first sight. Miller and Sanjurjo [6] (Appendices D, F.1 and F.2) provide all the technical details behind this bias for the interested reader. In brief, for $k = 1$ (which is the case we described in the example above), the streak selection bias can be thought of as a type of sampling without replacement effect. For $k > 1$ (which our example does not capture) there is an additional factor contributing to the streak selection bias namely, streak overlap. In fact, the restrictions imposed by the streak overlap on the selection of trials are even stronger.

We extend the permutation test to account for the non-identical nature of shots in an actual game setting. In particular, from an actual game we observe a sequence of shots for a player but not every shot has the same success probability. To estimate the make probability $p_i$ of a shot $s_i$ we use a dataset obtained from the SportVU optical tracking system for the 2013–14 and 2014–15 NBA seasons with approximately 200K shots each season. The dataset includes several variables that we use in our model:

- Distance to the basket

- Distance of the closest defender

- Player IDs for the shooter and the closest defender

- Touch time prior to shooting

- Number of dribbles before shooting

- Shot type (e.g., floater, tip etc.)

For our model, we build a feedforward neural network with 4 hidden layers. Each layer has 250 units and `relu` activation, i.e., $\text{ReLU}(y) = \max(0, y)$. We use a validation set for early stopping during training. Our final model has an out-of-sample accuracy of 66%, which is on par with the state-of-the-art shot make probability models [18]. We also evaluate the calibration of the output probabilities. The calibration is a measure of the accuracy of the predicted probabilities, rather than the pure—binary—outcome. Simply put, if there are two models $M_1$ and $M_2$ that predict a shot will be made with probabilities 51% and 85% respectively, and the shot is eventually made, while both have the same accuracy, they have different probability calibration; $M_2$ is more *certain* and better calibrated as compared to $M_1$. A typical way for evaluating the probability calibration of a classifier is through the reliability curve, that is, the plot of the predicted probabilities from the classifier versus the observed probabilities. In order to achieve this, and given that we cannot retake the exact same shot several times to observe how many times it was made, when obtaining the calibration curve we group the data points—in

our case shots—in bins based on their predicted probability. For example, using bins that cover a range of five percentage points of probability we bin together all the shots that were predicted by our model to have a shot make probability between $p$ and $p + 0.05$, with $0 \leq p \leq 0.95$. The average predicted probability over these shots is the *predicted* probability $p_{pred,b}$ for this bin $b$. We can then calculate the fraction of shots within this bin that were actually made, which constitutes the *observed probability* $p_{obs,b}$ for the bin. Ideally, for a well-calibrated model these two probabilities should be equal for every bin, i.e., $p_{pred,b} \approx p_{obs,b}$. Inset **(B)** at Fig 1 presents the reliability curve for our shot probability model on out-of-sample predictions. To obtain these predictions we perform 2-fold cross validation, where the 2 folds correspond naturally to the two seasons of shooting data we have. As we can see, our predicted probabilities follow closely the observed shot make probabilities.

We can now estimate the shot make probability for each shot in the data, which we will use for identifying the presence of "hot hand" or not. In order to make sure our predictions are out-of-sample, we perform a "leave-one-season-out" (LOSO) training. I.e., in order to make out-of-sample predictions for one season, we train the model on the rest of the seasons (in our case just a single other season). The neural network architecture used in each training is the same, and the out-of-sample performance of each LOSO model in terms of accuracy and calibration is statistically indistinguishable from the overall performance discussed above. LOSO ensures that there is no data leakage from the model training to our analysis and the estimation of the hot hand effect. We use players with at least 1000 shots during the two seasons. Given that there are 82 regular season games in each season, this means that we filter out from our analysis players that took approximately less than 6 shots per game. This threshold was chosen in order to provide, on average, sequences from individual games that can be used to examine the hot-hand hypothesis for values of $k > 1$. As we will see in our results, we are able to examine up to $k = 4$. Furthermore, for each player we only consider games for which we have information for all of their shots in that game. For example, in some cases the distance of the closest defender or the touch time is not provided. In this case we cannot estimate the shot make probability, and we filter out not only this specific shot, but all the shots from the same game, since the sequence will essentially be broken. This means that the total number of shots that we might have available for a player could be less than 1,000 despite the initial filtering.

The above process provides us with a sequence of $n$ shots $S \in \{M, X\}^n$ and an associated shot make probability vector $\mathbf{P} \in \Re^n$, which we term as the *shot history vector* where $p_i = \Pr[s_i = M]$. Given that the shots are not identical, we cannot simply permute the outcomes of the shots. However, we can repeatedly simulate (for our results we repeat these simulations 500 times each) the heterogeneous Bernoulli process describing the shot sequence to obtain $\Pr[M|\underbrace{M \ldots M}_{k \text{ times}}]_{sim}$ and examine whether there is enough statistical evidence to reject the null hypothesis of no hot hand. In this case the null translates to the probability of observing a make following $k$ consecutive makes in the data being the same with the one expected if we only considered the quality of the shot—i.e., independent of the outcome of the previous shot. We would like to emphasize here that in our analysis we do not simulate all the shots taken by a player and examine whether this generates simulated sequences with made shot streaks. Rather, we simulate and match shots that are part of a streak in the real data. This allows us to avoid potential systematic biases if for instance the situations in which shooters go on streaks are systematically different as compared to non-streak shots.

Despite the overall good performance of the model in terms of predicted probabilities (Fig 1(B)), every model is associated with an error. Therefore, any difference observed between $\Pr[M|\underbrace{M \ldots M}_{k \text{ times}}]_{sim}$ and $\Pr[M|\underbrace{M \ldots M}_{k \text{ times}}]_{data}$ will include this error as well. In order to adjust for

this, we also simulate for each player a random sequence of shots taken from the player and compare his expected from the model field goal percentage $\Pr[M]_{sim}$ with his actual field goal percentage $\Pr[M]_{data}$. This difference $\Pr[M]_{sim} - \Pr[M]_{data}$ provides an estimate of the error introduced by the model for this specific player and can be used to adjust the estimation of the hot hand effect size.

By applying the test described above on every individual player, we can identify whether there is statistical evidence supporting the hypothesis that they are "streaky shooters". However, given that we perform multiple tests—one per qualified player—the probability of obtaining a small number of false positive results for a given significance level $\alpha$ considered increases. As a robustness check, we can estimate the probability of a specific number of these tests returning positive results purely by chance. In particular, under the—realistic in our case—assumption that our tests are not correlated we can use the Binomial distribution for a meta-test. With $M$ tests each of which has a probability of $\alpha$ leading to a false positive result, we can estimate the probability of observing at least $r$ positive tests due to chance as: $\sum_{p=r}^{M} \binom{M}{p} \alpha^p (1-\alpha)^{M-p}$. If this probability is low, it consequently increases our confidence that the hot hand instances observed are not (all) false positives. This is indeed the case in our setting (see Results section).

## Results

For our results we start by estimating from game data the probability of a player $\pi$ making a shot $i$, conditioned on the previous shot they attempted (in the same game) being made as well $\Pr[M_{\pi,i}|M_{\pi,i-1}, M_{\pi,i-2}, \ldots M_{\pi,i-k}]_{data}$ for $k \in \{1, 2, 3, 4\}$ (to simplify and make the notation more compact, in what follows we will use $\Pr[M_{\pi}|\underbrace{M_{\pi} \ldots M_{\pi}}_{k \ times}]_{data}$ to refer to this probability).

We then compare this probability with the probability $\Pr[M_{\pi}|\underbrace{M_{\pi} \ldots M_{\pi}}_{k \ times}]_{sim}$ we should have

expected based on the quality of the shot(s) player $\pi$ took, and under the assumption that the shot sequence includes independent but not identically distributed shots. The difference $e_{\pi,k} = \Pr[M_{\pi}|\underbrace{M_{\pi} \ldots M_{\pi}}_{k \ times}]_{data} - \Pr[M_{\pi}|\underbrace{M_{\pi} \ldots M_{\pi}}_{k \ times}]_{sim}$ is the hot hand effect size for player $\pi$ and

for a sequence length of consecutive makes of length $k$. As described in the previous section, in order to estimate $\Pr[M_{\pi}|\underbrace{M_{\pi} \ldots M_{\pi}}_{k \ times}]_{sim}$ we develop a shot make probability model based on

a variety of contextual information (e.g., distance from the basket, distance of closest defender, shot type etc.). This sequence of shots is thus, a non-homogeneous Bernoulli process $B(n, \mathbf{P})$, where $\mathbf{P}$ is the shot history make probability vector for the sequence. Sampling this process for each player 100 times will provide us with the distribution for $\Pr[M_{\pi}|\underbrace{M_{\pi} \ldots M_{\pi}}_{k \ times}]_{sim}$. However, as aforementioned there is an error associated with the shot

make probability model, which will affect the estimation of $\Pr[M_{\pi}|\underbrace{M_{\pi} \ldots M_{\pi}}_{k \ times}]_{sim}$. To adjust for

this error we simulate a randomly selected set $S_{n,\pi}$ of player's $\pi$ shots—regardless of the success or not of his previous attempt(s)—and estimate the model error for $\pi$ as $\epsilon_{\pi} = \Pr[s_i = M]_{data} - \Pr[s_i = M]_{sim}, \ s_i \in S_{n,\pi}$. The adjusted hot hand effect size for player $\pi$ is then $\hat{e}_{\pi,k} = e_{\pi,k} - \epsilon_{\pi}$.

To reiterate, we use both seasons in our dataset, but when estimating the shot make probability history vectors we ensure these estimates are out-of-sample. Before looking all the results

**Table 1. Kemba Walker shows significant levels of streakiness, even after adjusting for errors in the shot make probability model.**

| $k$ | $\Pr[M\mid\underbrace{M\ldots M}_{k\ times}]_{data}$ | $\Pr[M\mid\underbrace{M\ldots M}_{k\ times}]_{sim}$ | $\Pr[s_i = M]_{data}$ | $\Pr[s_i = M]_{sim}$ | $\hat{e}_{KW,k}$ |
|---|---|---|---|---|---|
| 1 | 0.426 | 0.387 | 0.398 | 0.384 | 0.025*** |
| 2 | 0.471 | 0.388 | 0.401 | 0.385 | 0.067*** |
| 3 | 0.516 | 0.376 | 0.41 | 0.383 | 0.113*** |
| 4 | 0.533 | 0.388 | 0.410 | 0.392 | 0.127*** |

(p<0.1, *p<0.05, **p<0.01, ***p<0.001).

let us see examine the data for a specific player, Kemba Walker (KW), just as an illustration of the estimation process. We start by estimating the probability of KW making a shot he takes after consecutive makes of his last $k$ attempts (during the same game), i.e., $\Pr[M_{KW}\mid\underbrace{M_{KW}\ldots M_{KW}}_{k\ times}]_{data}$. We then simulate his shot sequence after $k$ consecutive makes 100 times to obtain his hot hand effect size $e_{\pi,KW}$. Table 1 shows the observed and expected from the shot make probability model field goal percentages after $k$ consecutive makes. As we can see KW performs better than expected after consecutive made shots. We then adjust for the model error, by simulating a random sample of KW's shots (these are *null* shots, i.e., they are not conditioned on the result of his previous shots) and comparing their observed and the expected field goal percentages as well. For each different value of $k$, the number of shots we sample is equal to the number of shots available for estimating $\Pr[M_{KW}\mid\underbrace{M_{KW}\ldots M_{KW}}_{k\ times}]_{data}$.

Given that for larger values of $k$ the sample size is significantly reduced, we want our model error estimate to account for the underlying uncertainty. For reference, for $k = 1$ our dataset includes 333 shots that KW took after a single make, while the sample size for the other values of $k$ is 138, 62 and 30 shots respectively. While the estimates from the null shots samples for the error model are closest to the observed field goal percentage for $k = 1$, for all values of $k$ the model underestimates Walker's probability of making a shot. Therefore, if not adjusting for the model error would lead to overestimation of his hot hand effect. Finally, the last column at Table 1 presents the adjusted hot hand effect size for KW, where the p-values correspond to a one-sided t-test with $H_0 : \hat{e}_{KW,k} = 0$, $H_1 : \hat{e}_{KW,k} > 0$. As we can see Kemba Walker appears to be "heating up" more, with the more shots he makes. We provide a detailed toy-example on the procedure of the introduced statistical test in "S1 Text". We also provide a discussion and additional results for the shots to be sampled for estimating the adjustment of the model errors in "S2 Text".

We performed the same process for all 153 players that took at least 1,000 shots over the two seasons covered by our data. Table 2 presents the league-wide results from the Binomial meta-test described in the previous section (at the 5% significance level). As we can see, for all values of $k$ our analysis provides at least 24 players that exhibit the hot hand effect. The probability of getting at least as many false positives is less than 1-in-1,000,000. Simply put, we are confident that these are true positives for the majority. The average adjusted hot hand effect size weighted by the sample size available for each player, is also shown at the same table. As we can see, it ranges from approximately 1.5% for $k = 1$, to about 5.8% for $k = 4$. In general, it appears that the effect size increase with the number of consecutive makes. However, here we have to put these results into context. For $k = 4$ the average sequence length we have for each player is around 33 shots. Thus, our test can be underpowered and unable to identify small hot

**Table 2. While overall the league exhibits shooting regression (last column), there are players that have the hot hand (HH) to different degrees (second column).** Out of the 153 qualified players for our statistical tests at least 24 players provided statistical evidence for streakiness at the 5% significance level for all values of $k$ examined. The probability of obtaining at least 24 false positives at the 5% level is less than 1-in-1,000,000.

| $k$ | # HH players | adj HH effect size | mean sequence length | overall adj effect size |
|---|---|---|---|---|
| 1 | 24 | 0.015 | 480.5 | -0.014 |
| 2 | 30 | 0.027 | 199.8 | -0.019 |
| 3 | 29 | 0.044 | 81.9 | -0.027 |
| 4 | 41 | 0.058 | 32.6 | -0.022 |

hand effects. Therefore, the average adjusted effect size among the players that exhibit the hot hand can be *inflated*.

We also estimate the average adjusted effect size over all 153 players, regardless of whether they provided evidence of the hot hand or not (last column at Table 2). As we can see this effect size is negative! In essence, overall as a league, players regress to their mean/expected shooting percentages after consecutive makes. Our take away from these results is that while shooting regression appears to be the stronger force over the whole league, there are individual players that exhibit statistically significant streakiness with varied effect size.

In order to further examine the importance of removing the assumption of identically distributed trials from the analysis, we compute the simulated probabilities by simply permuting the shots taken by a player within a game. This is essentially the same test as introduced in [6] to account for the streak selection bias. When ignoring the shot quality none of the qualified players exhibits the hot hand! This further supports our hypothesis that using statistical tests that rely on the identically distributed assumption can lead to severe underestimation of the phenomenon in the wild. We would like to emphasize here that our objective with this experiment is not to compare the conclusions obtained from the two methods, but rather to show the importance of considering the probability of each individual trial, when this is not constant. The permutation test described Miller and Sanjurjo [6] is appropriate whenever examining a series of trials that are identically distributed (e.g., free throws or three point contest shots). Furthermore, in the studies by Miller and Sanjurjo on controlled shooting (from three-point contests [20] and a shooting field experiment [21]) they also consider sequences of consecutive makes that might span different competition rounds, and, perform permutations across the different sessions/rounds for a player. This permutation eliminates any possible bias introduced by systematic variation between rounds, irrelevant to the hot hand, while it also eliminates any (partial) hot hand effect that is activated across sessions. Stratifying the data to more granular groups, was shown to reduce the statistical power of the permutation test, which in our case essentially means that we might be underestimating the phenomenon since we are using one such granular group (i.e., individual games).

## Discussion

We do not anticipate that our study will end the hot hand debate—and nor is it our goal—but we hope that it will lead to further studies of the quantification of the effect during in-game situations. Our results from two seasons of shooting data indicate that overall the league is subject to shooting regression. Regression to the mean [22] refers to the phenomenon of a future sample point of a random variable is likely to be closer to the mean of the process if its current sample point is extreme. In our setting, this means that players who perform above expectation over consecutive shots (i.e., $k$ makes), are likely to revert towards their shooting average, which, means they are likely to perform below expectation in terms of the shot make

probability. However, there are players that exhibit strong statistical evidence for the presence of the hot hand individually. An important context that we have to add in the hot hand analysis in actual game situations is that the presence of hot hand does not necessarily have to do with what fans might have in mind when they talk about a "player getting hot". It can be simply the ability of specific players to hunt and exploit good matchups for them within a game, leading to a streak of successful shots.

While not the focus of our study, identifying the underlying mechanisms that might be responsible for the phenomenon is the natural next question to ask. For example, can we facilitate our preparation for performing in order to trigger this streak? Is there a physical or mental mechanism that can (predictably) lead to a shooting streak in basketball? Is this streak a result of short-term neuroplasticity? Can focus and mental preparation potentially trigger this mechanism? Or is the majority of the effect due to players being able to exploit missmatches as alluded to above? These are question that while not possible to be answered by the observational data we have from success/failure sequences are crucial if we were to gain actionable insights for streaks of success. Furthermore, as far fetched as these hypotheses might sound, they can be plausible if streaks are not random permutations of binary outcomes. Understanding streak patterns (or lack thereof) can have important implications in decision making in areas beyond sports including investing, trading and purchasing behavior [23, 24]. Of course, we expect that these mechanisms—if existent—will be different for different areas and the appropriate context needs to be considered as well.

One of the main contributions of this study is the elimination of the assumption that trials within a sequence need to be identical. This is crucial particularly if one wants to study the streaks of success in a natural, not lab-like, environment. Beyond sports, existing literature has explored streaks in career trajectories and professional success [11]. These studies still make the implicit assumption that each career "trial" has the same probability of success. For example, when exploring the success of a director's movies not all movies have the same probability of being hits. A variety of factors ranging from cast and budget to the timing of its release [25] can factor in the success of a movie. The framework introduced in this report is applicable in a general setting, as long as, there is a model for the probability of success of a trial given appropriate contextual information. More importantly we include in our estimation an adjustment for the error associated with the model for the probability of success of a single trial.

It is worth noting that while the majority of the hot hand literature uses approaches based on combinatorics of binary sequences (such as runs tests, recently permutations tests etc.), regression analysis has also been used to study the phenomenon (e.g., [12, 26, 27]). In particular, Bocskocsky *et al.* [27] deal with the hot hand in NBA games as well and build a regression model to estimate the probability of scoring the next shot including as a covariate the player's "heat". The latter is defined as the difference between the actual and the expected success rate over the past $k$ shots. This means that a player can be "hot" (to some degree) even if he missed half of his last $k$ shots. Many might argue that this is not a hot player on a streak, and while there is not a right or wrong way to define what constitutes a hot hand, we choose to build a framework applicable on success/failure sequences, which is the typical setting in the hot hand literature. Furthermore, from a technical standpoint, the ordinary least squares regression was used in that study to model a binary variable—instead of the more appropriate logistic regression. This violates the constant variance assumption of linear regression. Consequently, this means that the t-statistic might not truly follow a t-distribution, leading to biased estimation of the corresponding p-values. Closer to our work is the study by Lantis and Nesson [12], where they examine various specifications of a regression model, with the dependent variable being an indicator on whether the shot examined was made, while the independent variables includes the outcome (make/miss) of the previous shot attempt, as well as, variables that

capture the difficulty of the shot. Their overall conclusion is that "making a streak of field goals decreases the probability that a player makes his next field goal" [12]. Note that the analysis considers the pooled data from all players, and hence, are comparable and in agreement with our league-wide results. Furthermore, the authors specifically state that not identifying an effect in the pooled data, does not preclude the possibility for individual players to exhibit evidence that the hot hand is not a fallacy (for them). In fact, our analysis at the individual player level shows that this is indeed the case.

It should be clear that the shot make probability model is crucial for studying the hot hand in game situations. These models can be improved by using information for the actual trajectory post-release from the shooter. In particular, Daly-Grafstein and Bornn [18] make use of features such as, entry angle to the hoop and entry location in the hoop that improve the prediction performance. However, these features are not available to us, and more importantly, even if they were they would not be particularly useful for our objective. Our goal is to estimate the shot make probability at the moment of release. Of course, these features are extremely useful as shown in [18] for estimating the shooting ability of players with fewer observations. Another potential problem in our setting is the possible changes (improvement or decline) in the shooting ability of players across seasons. While this is certainly a possibility across several seasons, we do not expect there to be a significant systematic bias across back-to-back seasons. For example, we calculated the year-to-year correlation in our data for the field goal percentage of the players with at least 1,000 shots over the two seasons to be equal to 0.76. Of course, the field goal percentage is a combination of both skill and shot selection. Nevertheless, this high levels of correlation point towards large changes in the field goal percentage from year-to-year being mainly outliers. Hence, we do not anticipate this to significantly or systematically affect the conclusions of the hot hand analysis.

In conclusion, while we clearly are not the first—or the last—to study the topic of hot hand, we believe that our report contributes to the existing literature in following ways. First, building on Miller's and Sanjurjo work [6] we introduce an empirical hypothesis that can analyze binary sequences of independent but not identically distributed trials. Second, we account for model errors associated with the estimation of the success probability for a trial. More importantly, our results are focused on a setting that has not been examined widely before (i.e., actual game situations).

## Supporting information

**S1 Text. A roadmap/example for applying our statistical test.**
(PDF)

**S2 Text. Model error adjustments.**
(PDF)

## Author Contributions

**Conceptualization:** Konstantinos Pelechrinis, Wayne Winston.

**Data curation:** Konstantinos Pelechrinis.

**Formal analysis:** Konstantinos Pelechrinis, Wayne Winston.

**Investigation:** Konstantinos Pelechrinis.

**Methodology:** Konstantinos Pelechrinis, Wayne Winston.

**Software:** Konstantinos Pelechrinis.

**Visualization:** Konstantinos Pelechrinis.

**Writing – original draft:** Konstantinos Pelechrinis, Wayne Winston.

**Writing – review & editing:** Konstantinos Pelechrinis, Wayne Winston.

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
