## [Decision Letter · Decision Letter 0]

17 May 2021

PONE-D-21-07619

The Hot Hand in the Wild

PLOS ONE

Dear Dr. Pelechrinis,

Thank you for submitting your manuscript to PLOS ONE. After careful consideration, we feel that it has merit but does not fully meet PLOS ONE’s publication criteria as it currently stands. Therefore, we invite you to submit a revised version of the manuscript that addresses the points raised during the review process.

You will find below (also one of the attached) the reports of two reviewers that are very familiar with the topic. As you will see they see promise in the paper and make a number of comments. Both of them ask for more data. As they said if you feel that you cannot analyze more data you need to explain the reason. Please note that I will send back the revised paper to the same reviewers. 

We look forward to receiving your revised manuscript.

Kind regards,

Pablo Brañas-Garza, PhD Economics

Academic Editor

PLOS ONE

Journal Requirements:

2. Please modify the title to ensure that it is meeting PLOS’ guidelines (https://journals.plos.org/plosone/s/submission-guidelines#loc-title). In particular, the title should be "specific, descriptive, concise, and comprehensible to readers outside the field" and in this case we feel that it is not informative and specific about your study's scope and methodology.

Reviewers' comments:

Reviewer's Responses to Questions

**Comments to the Author**

1. Is the manuscript technically sound, and do the data support the conclusions?

Reviewer #1: Yes

Reviewer #2: Yes

2. Has the statistical analysis been performed appropriately and rigorously? 

Reviewer #1: Yes

Reviewer #2: Yes

3. Have the authors made all data underlying the findings in their manuscript fully available?

Reviewer #1: Yes

Reviewer #2: Yes

4. Is the manuscript presented in an intelligible fashion and written in standard English?

Reviewer #1: Yes

Reviewer #2: Yes

5. Review Comments to the Author

Reviewer #1: The paper tests the "hot hand" effect using a data set from NBA shots but, unlike previous studies, relaxes the assumption that a player's shots are identically distributed. Instead, the paper uses information about each shot, such as distance to the basket, distance to the closest defender, touch time prior to shooting, or shot type, to construct a probability distribution about the chances that the current shot will be successful (a "make"). The authors then compare this probability against the observed probability, showing that the hot hand effect exists.

Main comments

1.The paper considers only two NBA seasons. Are they robust to data from more seasons, or are the results stronger in some seasons but weaker in other seasons? Given the applied nature of this paper, it could help the authors show that the results remain across multiple seasons.

2.From my interpretation, the authors calculate the probability distribution of a given shot being successful using the list of information in the bullet points of page 3 for each shot. What is a bit unclear is whether the probability of each shot being successful is drawn from this probability distribution (breaking the identically distributed assumption from other papers), whether they just use this probability to compare it against the observed probability that the shot was successful, or whether the authors do both. Please highlight which of these points the paper makes, as that will help differentiate it from the literature and, importantly, it will also help you convey the results of the paper to researchers in other fields.

3.It would be interesting if the authors can test their model in a free throw data set. For a given player, his shots may not be identically distributed (with the probability of "make" in each shot being drawn from a different distribution, or alternatively, the conditional probability of "make" shot k is a function of the number of previous "makes" in the free throw contest.

4.In the last paragraphs of page 5, you mentioned that you also condition the probability of a make on whether the player made the last two shots as well, showing that your results are qualitatively unchanged (although the intensity of the hot hand effect is diminished). It would be interesting, as a robustness check, if you can condition this probability on whether the player made a longer history of previous shots as well. Besides being a standard robustness check, it could help the reader understand how persistent the hot hand effect is, once a player starts experiencing it.

5.In the Discussion section, you mention several questions in the second paragraph, without offering some answers, or educated guesses, based on your results. From your findings, one could interpret that altering some of the independent variables listed in the bullet points of page 3 will affect the cumulative distribution function from where a player draws his probability of success in his next shot, ultimately affecting the emergence of the hot hand effect. If this interpretation is correct, then the authors should provide a clearer description of which variables listed in the bullet points of page 3 have the largest effect at the cumulative distribution function they construct. A clear understanding of these effect would provide more concrete policy implications, namely, which variables should a player (or a coach) affect to increase the chances of makes in each minute of a game.

Minor comments

1.Please define "relu" on page 3.

2.Please rewrite the paragraph at the bottom of page 3, as it's rather unclear.

3.Vector P is described in page 4 as "the vector with the shot make probabilities for each trial in the sequence". There is other, quite long, description of vector P in other sections of the paper. To simplify the notation, and clarify the intuition behind this vector, the authors could use notation from repeated games, calling P the "shot history."

4.The sentences starting at "Furthermore, from a technical standpoint, an ordinary least squares regression…" until "…estimation of the corresponding p-values" in page 7 are quite unclear. Please rewrite them to improve clarity. Are you refereeing to other papers in the literature using ordinary least squares in their regressions (that's what I think you mean) or did you use ordinary least squares at some point of your analysis?

Reviewer #2: The authors use two full seasons of shot outcome and shot situation game data from SportVU optical tracking system. They use the first year to train a model of shot probability for different types of shots taken throughout the course of a game, for each particular player, accounting for things like distance to the basket and to the closest defender. Then they use these estimated shot probabilities to simulate shot sequences for each player in the 2nd year, based on the types of shots they take, out of sample, and use these simulated datasets to construct null distributions for their stat tests of shooting performance in the second year.

The paper has promise, but needs a bit more work, as I outline below.

Major Comments:

1. The authors' analysis is limited to two seasons of NBA data, which results in shot sample sizes for individual players that the authors state are too small to test for shooting performance on streak lengths longer than two. They offer no explanation for why they did not consider more years of data. Using more data would be nice for a number of reasons. For one, the tests would be more powered on the individual level. Second, they could consider performance on streaks of length three, which is fairly standard in the literature. Third, they could consider more individual shooters than the 21 that they currently consider (they should explain how they selected the 1000 shot inclusion criterion, and the sensitivity of their results to it). This would make their tests more robust, and the selection of shooters would be more representative of the typical shooter in the league. I hope that the authors can obtain more data and extend their analysis to the larger set of data.

2. As mentioned above, the authors train their model of hit probability for each shot type for each player using one season (the first) then use this to build null distributions for testing shot performance of the players in the second season. I understand that this may be standard procedure for "hold-out" out-of-sample testing, but I don't understand why the authors do not perform robustness checks. For example, cross-validation makes sense to me here, starting with instead using the second year to train the model, then test on the first year of shooting performance. Also, is it overly problematic to train the model on both years of the data then test it on performance from both periods (at least for a robustness check)? In principle, this would seem to allow for a more apples-to-apples analysis given that the model would be better calibrated to the "true shot probabilities." Also, this would allow more shot data to be used in tests on performance. Though as mentioned in comment 1, I hope the authors can obtain and analyze more data.

Other Comments:

1. The authors quickly mention that if they were to have instead run a permutation test on shooting performance when on streaks of hits, permuting on the game x individual level, then they would have found much less evidence of streak shooting. This is a bit misleading in the sense that permuting on the game level eliminates the possibility of detecting any hot hand that initiates between (rather than within) games, whereas the authors' primary analysis does not. Thus, this is a bit of a stacked comparison. In Miller and Sanjurjo's working papers on controlled shooting experiments (R&R at Review of Economics and Statistics) and the NBA Three Point Contest (forthcoming at European Economic Review) they perform robustness checks in which they consider more granular permutation strata, e.g. for contest year, shooting round, ball on rack within round, and so on. Stratifying on the "contest x round" level (which would be the most similar to permuting on the game level in the authors work, tends to reduce statistical significance, for the reasons they discuss: it desensitizes tests to both: hot hand that activates between rounds, as well as other systematic changes in shooting behavior that activates between rounds not due to the hot hand. In this sense it is conservative to permute on the more granular levels. The authors should qualify their discussion accordingly.

2. It seems it would be worth adding a bit of discussion on the variables the authors use vs. those used by Bockocksky et al, and Rao before them, and explaining why there are differences, if there are.

3. There are a few things the authors can clean up: (i) the discussion of base rate vs. p(M|M)_perm vs. p(M|M)_data can be written better; as is it is a bit confusing, (ii) The discussion on the stability of effect size for streaks of length one vs two is potentially a bit misleading; depending on the model of hot hand shooting, the extent of hot hand expected on each of these two shot situations could easily be different, and there is an attenuation bias in true effect size due to measurement error (which varies with streak length) that is pointed out in the literature in work by Arkes, Stone and Arkes, and in each of the papers of Miller and Sanjurjo, (iii) the second para of the intro is written as if GVT were unaware of potential confounds in game data; this is misleading in the sense that they acknowledge this so consider also free throw shooting and conduct a controlled shooting experiment, (iv) similarly, say "free throw attempts or three-point contests are typically used when studying the phenomenon in basketball"; here, controlled shooting studies are excluded; the authors should consider citing the papers by koehler and Conley and Miller and Sanjurjo on 3pt shooting contest, and the controlled shooting study of Gilovich et al, and the analysis of several controlled shooting studies in another paper by Miller and Sanjurjo. Similarly, this may be the place to quickly cite other work on NBA game shooting, (v) (last para of the Intro) the authors state that permutation tests are common in hot hand studies with basketball data (and in discussion say "permutation tests have been used by the majority of the hot hand literature.."). They are used in Miller and Sanjurjo's work, but were not used in GVT and those that followed, until the recent work; in particular, who has used permutation tests on game data, as the authors suggest?, (vi) in the same paragraph the authors suggest that permutation tests are vulnerable to the small sample bias observed in Miller and Sanjurjo (2018), but those authors make clear that permutation tests under the i.i.d. assumption are not vulnerable to the bias. The authors should make clear what biases they are referring to. As written "b" does not seem correct to me. As written it seems possible that GVT conducted permutation tests that were vulnerable to the small sample bias. This is not the case, (vii) the small sample bias does not just appear in small samples; it appears in all samples but is more pronounced in small samples.

4. The writing can be polished a bit. For example, "extent" rather than "extend", "shot" not "show", "sampling this process...several times" should be more explicit, e.g. 10,000 times, I´m not sure "robust" is the right word when talking about whether the hot hand is common across players, "in the different.", "is less than 1&", "decisions making"

6. PLOS authors have the option to publish the peer review history of their article (what does this mean?). If published, this will include your full peer review and any attached files.

Reviewer #1: No

Reviewer #2: No

---

## [Author Response · Author response to Decision Letter 0]

21 Jul 2021

We would like to thank the reviewers for the useful comments. These comments have helped us to significantly improve - in our opinion - the quality of the manuscript. The additional results obtained as a consequence of the required changes have allowed us to refine our conclusions. The major changes we have made - among several changes detailed in our answers below and marked with red fonts in the revised manuscript - can be summarized in the following: 

We extended our analysis with data from the 2013-14 season. Thus, we now have 2 seasons of shots to run our analysis. In particular, in order to obtain the shot make probability for the shots during each season we train the model on the data from the other season. This ensures that the probabilities used are estimated out-of-sample. Then for each player we use their shot data from both seasons. 

We have added an adjustment at the estimation of the hot hand effect size to account for the errors introduced from the shot make probability model. In particular, the shot make probability is associated with some uncertainty/error for which we adjust. 

Following are our detailed responses to each reviewer’s comments/questions. 

Reviewer 1

1.The paper considers only two NBA seasons. Are they robust to data from more seasons, or are the results stronger in some seasons but weaker in other seasons? Given the applied nature of this paper, it could help the authors show that the results remain across multiple seasons.

We would like to thank the reviewer for this comment and we agree that an analysis including more seasons would add robustness in the observed results. However, unfortunately we only have 2 years of tracking data for the shots. While it is possible to obtain shot locations and outcomes for multiple years from play-by-play data, these do not include crucial variables like closest defender distance, touch time before the shot, dribbles prior to the shot, defender ID etc. Therefore the quality of the shot - i.e., the make probability for a shot - cannot be assessed accurately. Nevertheless, in the first version of the manuscript we trained the shot make probability model on the data from the 2013-14 season and then run the “hot hand” analysis on the 2014-15 season in order for the predictions from the model to be out-of-sample. In the revised version of the manuscript, we trained again the shot make probability model using the same neural network architecture on the data from the 2014-15 season and ran the “hot hand” analysis on the 2013-14 season as well. This way the predictions from the model are still out-of-sample, while we have now a larger observation set from players from 2 seasons. 

2.From my interpretation, the authors calculate the probability distribution of a given shot being successful using the list of information in the bullet points of page 3 for each shot. What is a bit unclear is whether the probability of each shot being successful is drawn from this probability distribution (breaking the identically distributed assumption from other papers), whether they just use this probability to compare it against the observed probability that the shot was successful, or whether the authors do both. Please highlight which of these points the paper makes, as that will help differentiate it from the literature and, importantly, it will also help you convey the results of the paper to researchers in other fields.

We would like to thank the reviewer for bringing up this issue, which allows us to clarify. We have better explained this part in the revised manuscript, but in brief, this model outputs a “make” probability for each shot in our dataset. We use this probability to simulate this same shot several times - i.e., through a “biased coin” according to the shot make probability - and compare the simulation results from a series of shots, with the actual outcome of the same series in our dataset. 

One possible point of confusion might be the use of the term “observed” probability for the calibration curve of the model. This is purely for evaluation purposes of the shot make probability model, and it is the standard approach and terminology for probabilistic model evaluations. In particular, we cannot know the true probability of a shot being made. However, we can group shots with the same/similar predicted probability and then obtain the observed probability as the percentage of these shots actually made. For example, if we take all the shots where the model predicts a make probability of 55%, then the observed probability is simply the percentage of these shots that were actually made during the games. We have clarified this point further in the revised manuscript. 

3.It would be interesting if the authors can test their model in a free throw data set. For a given player, his shots may not be identically distributed (with the probability of "make" in each shot being drawn from a different distribution, or alternatively, the conditional probability of "make" shot k is a function of the number of previous "makes" in the free throw contest.

We would like to thank the reviewer for this comment. This analysis is certainly possible, and FTs are one of the settings examined in the hot hand literature. While the tracking dataset we used in our manuscript does not include free throws, obtaining FTs is possible through play-by-play data. The challenge with FTs is that they are with high confidence identically distributed. Several different splits (e.g., home-away, first-second half etc.) provide a similar overall FT% for a player, allowing us to treat all FTs as identical trials. Thus, Miller’s and Sanjurjo permutation test is directly applicable here. We have run the permutation test on FT data from the 2019-20 season by randomly permuting FTs within the same game and using the sequence from all the games to estimate the probability Pr[Make FT_i | Make FT_{i-1}]. For example, LeBron James has a 71.8% probability of making a free throw after he made his previous one, which is higher compared to the 66.9% that we would expect at random - i.e., after permuting his FTs - with a p-value of 0.003. One the other hand, Steven Adams, does not exhibit the hot hand at FTs. He makes 52.3% of his FTs after a make, which is statistically indistinguishable (p-value 0.26) from the 49.2% expected from random permutations of his FTs. The results from the FTs are similar with the results from in-game shots, in the sense that there are players with strong effect and others with no effect. However, given that this analysis does not require removal of the assumption of identical trials, we decided to not include these results in the revised manuscript as we felt that it might break the flow of the study and confuse the reader in terms of the objective of the study, which is tied to the non-identical nature of the trials. 

4.In the last paragraphs of page 5, you mentioned that you also condition the probability of a make on whether the player made the last two shots as well, showing that your results are qualitatively unchanged (although the intensity of the hot hand effect is diminished). It would be interesting, as a robustness check, if you can condition this probability on whether the player made a longer history of previous shots as well. Besides being a standard robustness check, it could help the reader understand how persistent the hot hand effect is, once a player starts experiencing it.

This is a very interesting point and indeed being able to condition on long series of successes is going to provide additional insights on the persistence of the phenomenon. In the original manuscript, given that we used one season of data to analyze/quantify the hot hand phenomenon the sample sizes became very small for larger values of k. However, in the revised manuscript we have used two seasons of data and by reducing the constraints for players qualified for our analysis (in terms of total number of shots) we were able to examine values of consecutive makes up to 4. The results show that the phenomenon is persistent for individual players. We have added a detailed description of these results in the revised manuscript. 

5.In the Discussion section, you mention several questions in the second paragraph, without offering some answers, or educated guesses, based on your results. From your findings, one could interpret that altering some of the independent variables listed in the bullet points of page 3 will affect the cumulative distribution function from where a player draws his probability of success in his next shot, ultimately affecting the emergence of the hot hand effect. If this interpretation is correct, then the authors should provide a clearer description of which variables listed in the bullet points of page 3 have the largest effect at the cumulative distribution function they construct. A clear understanding of these effect would provide more concrete policy implications, namely, which variables should a player (or a coach) affect to increase the chances of makes in each minute of a game.

We would like to thank the reviewer for this comment. Our objective with posing these questions in the discussion is to point towards some directions for future research. Unfortunately, we are not really able to identify the underlying mechanisms (physiological, neurological or psychological) that lead to the emergence of the phenomenon for some athletes with the observational data of makes and misses. As we mention in the revised discussion the emergence of the hot hand phenomenon in game-situations might actually not even have to do anything with factors that come to our mind when we talk about the hot hand, such as psychological, but rather be a result of a player being able to hunt and exploit favorable matchups consistently during a game. While we can not provide answers to the questions described at this part of the manuscript, we believe it is useful to pose these questions in order to lay a set of potential paths for other researchers (as well as our own future work) to take on. We have re-edited that part of the manuscript to clarify our objective with these questions. 

Furthermore, the shot make probability model is certainly crucial for the results from our analysis. An inaccurate model will lead to over- or underestimation of the phenomenon, depending on whether the model under -or over-predicts the shot make probability respectively. In the latest manuscript we have revised our framework to include an adjustment for possible errors associated with the shot make probability model. 

An interesting, tangential to the reviewer's point here is the reaction of the players themselves to consecutive makes. As Bocskocsky et al. (2014) showed, when a player makes a shot they tend to take tougher shots next, i.e., shots with lower probability of success (e.g., shots further away from the basket, heavily contested etc.). However, our analysis adjusts for this, since the simulated makes/misses are drawn from the shot make probability model. 

Minor comments

1.Please define "relu" on page 3.

We have defined the relu function. 

2.Please rewrite the paragraph at the bottom of page 3, as it's rather unclear.

We have edited the paragraph, and we believe it is more clear now. 

3.Vector P is described in page 4 as "the vector with the shot make probabilities for each trial in the sequence". There is other, quite long, description of vector P in other sections of the paper. To simplify the notation, and clarify the intuition behind this vector, the authors could use notation from repeated games, calling P the "shot history."

We would like to thank the reviewer for this suggestion, which we have followed in the revised manuscript. 

4.The sentences starting at "Furthermore, from a technical standpoint, an ordinary least squares regression…" until "…estimation of the corresponding p-values" in page 7 are quite unclear. Please rewrite them to improve clarity. Are you refereeing to other papers in the literature using ordinary least squares in their regressions (that's what I think you mean) or did you use ordinary least squares at some point of your analysis?

We would like to thank the reviewer for this comment. We indeed mean that the prior literature cited has used OLS. We have updated this part of the manuscript to make it more clear. 

Reviewer 2

1. The authors' analysis is limited to two seasons of NBA data, which results in shot sample sizes for individual players that the authors state are too small to test for shooting performance on streak lengths longer than two. They offer no explanation for why they did not consider more years of data. Using more data would be nice for a number of reasons. For one, the tests would be more powered on the individual level. Second, they could consider performance on streaks of length three, which is fairly standard in the literature. Third, they could consider more individual shooters than the 21 that they currently consider (they should explain how they selected the 1000 shot inclusion criterion, and the sensitivity of their results to it). This would make their tests more robust, and the selection of shooters would be more representative of the typical shooter in the league. I hope that the authors can obtain more data and extend their analysis to the larger set of data.

We would like to thank the reviewer for this comment. We agree that having more data would help in several fronts as detailed in the comment. However, tracking data are publicly available only for the two seasons we have access to. The NBA has made this detailed information about shots private/proprietary after the 2014-15 season. Therefore, adding more data in our analysis is not possible. However, we updated our analysis in order to increase the sample size - both for individual players and in terms of players examined - and to allow us to perform some of the things recommended by the reviewer’s comment. In particular, in the original manuscript we only used the 2014-15 season for examining the hot hand. The thought process is that we will train (test and validate) the shot probability model with the 2013-14 season data and then evaluate the hot hand on an out-of-sample dataset. In the revised version of the manuscript we still evaluate the hot hand using out-of-sample data, but we trained two separate models using the same neural network architecture. The one is using the 2013-14 season’s data to learn the model weights and then make predictions for the shot make probability of the 2014-15 season shots, while the second model is trained on the data from the 2014-15 season and used to estimate the out-of-sample shot make probability for the 2013-14 season. This way there is no data leakage from the training of the model to the predictions for the shot making probability in the player/shot sequences of interest. 

Furthermore, we agree that our initial filtering process was too stringent. In the revised manuscript, we have loosened the threshold for the qualified players to be included in our analysis, by filtering out players with less than 1,000 shots over both seasons in the data. The reason for using this filter is that by allowing players that do not meet this criteria, we will end up including in our analysis players that took on average less than 6 shots per game. These sequences will not provide usable data, particularly for larger values of streak length. This filtering leads to 153 qualified players (as compared to the 21 players included in the first version of the manuscript), while it also allows us to examine streak lengths up to 4. 

2. As mentioned above, the authors train their model of hit probability for each shot type for each player using one season (the first) then use this to build null distributions for testing shot performance of the players in the second season. I understand that this may be standard procedure for "hold-out" out-of-sample testing, but I don't understand why the authors do not perform robustness checks. For example, cross-validation makes sense to me here, starting with instead using the second year to train the model, then test on the first year of shooting performance. Also, is it overly problematic to train the model on both years of the data then test it on performance from both periods (at least for a robustness check)? In principle, this would seem to allow for a more apples-to-apples analysis given that the model would be better calibrated to the "true shot probabilities." Also, this would allow more shot data to be used in tests on performance. Though as mentioned in comment 1, I hope the authors can obtain and analyze more data.

We would like to thank the reviewer for this comment - which is relevant to the first one. Using in-sample predictions could lead to understating the hot hand effect since the predicted probability would be biased towards the actual outcome of the shot (either made or missed). Now, of course the question is how much will this bias be, and the answer here is not trivial. After the reviewer’s comment we actually experimented with a leave-one-out setting (i.e., train a model with all the data, and one with all the data but one particular shot) and we found that the bias does not appear to be more than half percentage point in the majority of the cases examined. However, in our case this could still be a large bias relative to the effect size of the hot hand (approximately two percentage points). Therefore, we decided to stay with out-of-sample predictions for the hot hand. Nevertheless, as detailed in our answer on the first comment, we have now used more data to train the model. 

Moreover, while training the neural network we have used a validation set for early stopping. Finally, we performed some robustness checks by training the model on the 2013-14 (2014-15) season and evaluating on the 2014-15 (2013-14) season, and the performance in terms of accuracy, Brier score and validation curve is practically indistinguishable. In the revised manuscript, we have included the results from the updated process described in our response on the first comment. 

Other Comments:

1. The authors quickly mention that if they were to have instead run a permutation test on shooting performance when on streaks of hits, permuting on the game x individual level, then they would have found much less evidence of streak shooting. This is a bit misleading in the sense that permuting on the game level eliminates the possibility of detecting any hot hand that initiates between (rather than within) games, whereas the authors' primary analysis does not. Thus, this is a bit of a stacked comparison. In Miller and Sanjurjo's working papers on controlled shooting experiments (R&R at Review of Economics and Statistics) and the NBA Three Point Contest (forthcoming at European Economic Review) they perform robustness checks in which they consider more granular permutation strata, e.g. for contest year, shooting round, ball on rack within round, and so on. Stratifying on the "contest x round" level (which would be the most similar to permuting on the game level in the authors work, tends to reduce statistical significance, for the reasons they discuss: it desensitizes tests to both: hot hand that activates between rounds, as well as other systematic changes in shooting behavior that activates between rounds not due to the hot hand. In this sense it is conservative to permute on the more granular levels. The authors should qualify their discussion accordingly.

We would like to thank the reviewer for this thoughtful comment. Indeed, our analysis is only able to identify hot-hand effects within a game since we are not considering consecutive shots across games. While in a three-point contest examining streaks that span consecutive rounds (rather than only shots during the same round) makes a lot of sense -- similar to a game situation and halftime for example -- in game performance it does not seem to be particularly relevant. However, we agree that this approach would provide conservative estimates if there is hot-hand that activates between rounds/games. Nevertheless, when estimating the effect for individual players, we use the whole sequence of qualifying shots (i.e., a shot taken after k makes) with the constraint that all these k+1 shots need to be within the same game. This provides us with fairly large sequences/sample size (particularly for k = 1 and k = 2) that allows us to avoid statistical power issues for smaller sample sizes of binary data. 

Furthermore, we would like to clarify that in our uniform permutation test (i.e., without considering the shot quality), our goal was not to per se compare with Miller and Sanjurjo’s work directly in terms of the conclusions about the presence of hot hand or not, but rather to showcase that ignoring the shot quality can lead to underestimation of the effect. Again during this permutation we consider all qualified shots for a given player, but we permute only shots within the same game. We have added and clarified these points in the revised manuscript as per the recommendations of the reviewer. 

2. It seems it would be worth adding a bit of discussion on the variables the authors use vs. those used by Bockocksky et al, and Rao before them, and explaining why there are differences, if there are.

We have added some discussion on the similarities and differences between our shot make probability model and those from A. Bocskocsky et al., as well as, Daly-Grafstein and Bornn. In brief, compared to the model from Bocskocsky et al., there is quite a bit of overlap in terms of the variables used, but there is a clear difference in the modeling approach. On the other hand Daly-Grafstein and Bornn, make use of more granular tracking data, and in particular the actual trajectory of the ball after a shot, from which they extract variables such as the entry angle to the hoop, the distance from the center of the hoop etc. These variables are not available at the (public) dataset that we used. 

3. There are a few things the authors can clean up: (i) the discussion of base rate vs. p(M|M)_perm vs. p(M|M)_data can be written better; as is it is a bit confusing, (ii) The discussion on the stability of effect size for streaks of length one vs two is potentially a bit misleading; depending on the model of hot hand shooting, the extent of hot hand expected on each of these two shot situations could easily be different, and there is an attenuation bias in true effect size due to measurement error (which varies with streak length) that is pointed out in the literature in work by Arkes, Stone and Arkes, and in each of the papers of Miller and Sanjurjo, (iii) the second para of the intro is written as if GVT were unaware of potential confounds in game data; this is misleading in the sense that they acknowledge this so consider also free throw shooting and conduct a controlled shooting experiment, (iv) similarly, say "free throw attempts or three-point contests are typically used when studying the phenomenon in basketball"; here, controlled shooting studies are excluded; the authors should consider citing the papers by koehler and Conley and Miller and Sanjurjo on 3pt shooting contest, and the controlled shooting study of Gilovich et al, and the analysis of several controlled shooting studies in another paper by Miller and Sanjurjo. Similarly, this may be the place to quickly cite other work on NBA game shooting, (v) (last para of the Intro) the authors state that permutation tests are common in hot hand studies with basketball data (and in discussion say "permutation tests have been used by the majority of the hot hand literature.."). They are used in Miller and Sanjurjo's work, but were not used in GVT and those that followed, until the recent work; in particular, who has used permutation tests on game data, as the authors suggest?, (vi) in the same paragraph the authors suggest that permutation tests are vulnerable to the small sample bias observed in Miller and Sanjurjo (2018), but those authors make clear that permutation tests under the i.i.d. assumption are not vulnerable to the bias. The authors should make clear what biases they are referring to. As written "b" does not seem correct to me. As written it seems possible that GVT conducted permutation tests that were vulnerable to the small sample bias. This is not the case, (vii) the small sample bias does not just appear in small samples; it appears in all samples but is more pronounced in small samples.

We would like to thank the reviewer for this detailed comment on a few misrepresentations we had in the first version of the manuscript. We have updated the revised manuscript accordingly and in particular: 

(i) We have updated the discussion on our simulations/permutations 

(ii) This is a particularly important issue that we thought more about after the reviewer’s comment and led to one of the major changes in the revised manuscript. In particular, and as detailed in our responses above, we have added a mechanism for adjusting for the shot make probability model’s errors with a detailed description in the revised manuscript.

(iii) This was certainly not what we wanted to imply. However, we see now how this could be perceived this way. We have re-written that part of the introduction to clarify. 

(iv) We have included studies on controlled shooting, which we missed in our initial version

(v) We erroneously used the term “permutation tests” in this part of the paper. We were referring in general to analysis of binary sequences that can include permutation tests, runs tests etc. We have changed the wording at this part to clarify that we are talking about combinatorics of binary sequences in general. 

(vi) We have re-written this part to clarify our point. Our point was not that the permutations tests are vulnerable to the small sample bias observed by MS. We were referring to biases from assumptions that each trial is identical (i.e., same probability of success). We have updated the text accordingly 

(vii) We have also clarified this point. 

4. The writing can be polished a bit. For example, "extent" rather than "extend", "shot" not "show", "sampling this process...several times" should be more explicit, e.g. 10,000 times, I´m not sure "robust" is the right word when talking about whether the hot hand is common across players, "in the different.", "is less than 1&", "decisions making"

We have made a careful editing of the revised manuscript, fixing a lot of the typos and grammatical errors in the first version.

---

## [Decision Letter · Decision Letter 1]

10 Aug 2021

PONE-D-21-07619R1

The Hot Hand in the Wild

PLOS ONE

Dear Dr. Pelechrinis,

Thank you for submitting your manuscript to PLOS ONE. After careful consideration, we feel that it has merit but does not fully meet PLOS ONE’s publication criteria as it currently stands. Therefore, we invite you to submit a revised version of the manuscript that addresses the points raised during the review process.

You will find the reports attached. As you will see Reviewer 1 is entirely satisfied while Reviewer 2 is not. His report is extensive and specific and personally I feel that the paper will be clearer following his advice. On top of that he notes that the paper needs to tone down the contribution and putting into the context of the existing literature.

We look forward to receiving your revised manuscript.

Kind regards,

Pablo Brañas-Garza, PhD Economics

Academic Editor

PLOS ONE

Journal Requirements:

Additional Editor Comments (if provided):

Reviewers' comments:

Reviewer's Responses to Questions

**Comments to the Author**

1. If the authors have adequately addressed your comments raised in a previous round of review and you feel that this manuscript is now acceptable for publication, you may indicate that here to bypass the “Comments to the Author” section, enter your conflict of interest statement in the “Confidential to Editor” section, and submit your "Accept" recommendation.

Reviewer #1: All comments have been addressed

Reviewer #2: (No Response)

2. Is the manuscript technically sound, and do the data support the conclusions?

Reviewer #1: Yes

Reviewer #2: Partly

3. Has the statistical analysis been performed appropriately and rigorously? 

Reviewer #1: Yes

Reviewer #2: N/A

4. Have the authors made all data underlying the findings in their manuscript fully available?

Reviewer #1: Yes

Reviewer #2: Yes

5. Is the manuscript presented in an intelligible fashion and written in standard English?

Reviewer #1: Yes

Reviewer #2: Yes

6. Review Comments to the Author

Reviewer #1: The author addressed all my comments, both major and minor. The specific contribution of the paper is now clearer and the explanations easier to follow for a general audience.

Reviewer #2: The authors effectively double their dataset by performing the same analysis from first year to second, and now from second to first also. In addition, the peform an adjustment for measurement error in their model. Together, with these results the pattern of results changes qualitatively.

I think the paper has promise, but still needs quite a bit of work to be publishable, as I explain below.

Main Points:

1. The authors need to do a much better job of relating their work to the previous literature. For example, their work is quite closely related to that of Rao (2009) and Lantis and Nesson (2019). Neither of these papers is cited.

Lantis and Nesson (2019) study 12 years of NBA shooting data, including the 2 years studied by the current authors. In addition, they study both field goal shooting and free throws, as opposed to just field goal shooting. Further, they use a considerably longer list of controls than the current authors.

Their pooled test results are similar, and they don't have tests on the individual level like the current authors now do.

The authors cite Bocskocsky et al. (2014) as if it is perhaps the only regression analysis that has been performed in the hot hand fallacy literature. This is very misleading. Arkes, Lantis and Nesson, Green and Zweibel, Miller and Sanjurjo, and others use regression analysis as well.

The authors should make this clear. It is a valid, and typical, approach used in the literature. Multiple approaches have been taken to correct for the bias pointed out by Miller and Sanjurjo. The particular weaknesses of Bocskocksky et al. are not general to these studies.

An obvious question is what result would these other typical regression approaches produce with the same data. Lantis and Nesson provides an answer to this question, with more data.

2. If I understood the authors' empirical approach correctly, they are comparing the computed P(M|M...M)_sim in (heterogenous Bernoulli) simulated datasets to the observed P(M|M...M)_data in the real data, rather than comparing P(M|M...M)_data to performance on the directly matched shots taken on the same streaks the real shooter was actually on. The latter approach seems more direct, and that it will be less suceptible to measurement error. The former approach may even be suceptible to systematic bias if the types of situations in which real shooters go on streaks create systematic differences in the sampling of their streak shots and those in the simulated datasets.

I suggest that the authors run the analysis I mention, and compare results with those of their current analysis, or if they agree that it is better, replace the previous analysis with it. (Later, I see new added text "We then simulate his shot sequence after k consecutive makes 100 times..."; this suggests the latter approach; this should all be explained much clearer, so there is no ambiguity)

3. Relatedly, in the revision the authors apply an ad hoc model error adjustment that samples from all shots (not just streak shot situations), then apply it to adjust hot hand estimates. First, this is a potentially serious issue with the modeling approach that the authors should address in a more complete way. Why is this happening? Is it happening systematically across shooters? If so, why? Second, if this adjustment samples at random from all shots (including shots taken on streaks) then this is creating in turn another mechanical bias against hot hand effect size. An alternative would be to sample without including these shots, which is not ideal for other reasons, but perhaps better. In either case, because this adjustment is non-trivial in size, if the authors are going to use it, they should make sure that it is not biasing hot hand effect sizes in some other more subtle way. For example, they can run simulations using different models of hot hand shooting (that is, in which they have assumed there are hot hand shooters), and see if their model systematically produces errors in the same direction. If so, this would suggest that the measurement error they are picking up is actually a mechanical artifact of hot hand shooting, so making the correction they are could bias results against finding the hot hand.

Are the errors similar in both directions when the authors train the model in one direction, then the other?

4. in the KW example how is it possible that he only has 333 shots following at least one make, if he has taken more than 1000 shots? The only explanation that seems to make sense is that the authors are considering streaks of exactly one hit, rather than of at least one hit. But conditioning on at least one hit makes more sense because it provides more powered tests. The same goes for longer streak lengths.

5. The authors should discuss how using one year to train a model to make probability estimates for another year can affect their results if there are systematic differences in shooting performance from one year to the next, e.g. the shooter improves, or changes technique over the offseason.

6. The sampling without replacement example to explain the bias discovered by Miller and Sanjurjo is not correct. In the Econometrica they show that the bias is generally larger than the analogous sampling without replacement type effect.

7. In response to point 3 of the other reviewer, the authors argue that an analysis of free throw shooting is not interesting for them to include because--unlike field goals---free throws are essentially identically distributed. This is not correct. Players shoot systematically better on the second ft than the first, for example.

8. The authors say:

"In particular, we group the shots in our test set based on their predicted probability, and for each set, we estimate the fraction of them that were actually made."

I am confused. What grouping? This should be explained clearly from the outset.

Other Comments:

1. It is not that the streak selection bias "could lead to underestimation of the hot hand phenomenon", rather that it does lead to the underestimation.

2. The authors say:

"Inset (B) at Figure 1 presents the reliability curve for our shot probability model as obtained through an (out-of-sample) test set."

Which one? Why just one?

3. Need to explicitly state how many simulations of the heterogenous Bernoulli process were performed. "...repeatedly simulate" is too vague. Also the statistical testing procedure should be explained clearly.

4. The authors say:

"We use players with at least 1000 shots during the two seasons. Given that there are 82 regular season games in each season, this means that we filter out from our analysis players that took approximately less than 6 shots per game. This threshold was chosen in order to provide, on average, sequences from individual games that can be used to examine the hot-hand hypothesis for values of k > 1. As we will see in our results, we are able to examine up to k = 4."

What is this claim based on? It needs to be backed up by some evidence. Presumably the authors performed a power analysis, and they are summarizing the results here. They need to explain the results of the power analysis in a clearer way. They also suggest elsewhere that they are sufficiently powered. Based on what analysis?

5. One cannot "calculate the probability" of Kemba Walker making a shot. Should fix the language.

6. The authors say:

"Furthermore, in the studies by Miller and Sanjurjo on controlled shooting (both actual three-point contests [19] and a shooting field experiment [20]) they also examine whether there is hot hand activated between the different contest rounds. They achieve this by considering sequences of consecutive makes that might span different competition rounds."

This is not quite representative of Miller and Sanjurjo's approach. They first permute on the player level, to facilitate comparison with the previous literature's results, then on the player x session (or round) level, which does two things. First it eliminates vulnerability in the estimates to any systematic variation between sessions (not due to the hot hand), but if any hot hand effect at least partially activates between sessions or rounds it eliminates that too. By contrast, they are not actively trying to "examine whether there is a hot hand activated between the different contest rounds."

7. The authors say:

"Our results from two seasons of shooting data indicate that overall the league is subject to shooting regression, i.e., players shoot below expectation after consecutive makes, thus, regressing towards their shooting average. However, there are

players that exhibit strong statistical evidence for the presence of the hot hand individually."

This writing is not clear. First, the first statment is about the average or representative shot, across shooters. I don't understand the use of "regression towards their shooting average" here. Which shooting average?

8. The authors say:

"An important context that we have to add in the hot hand analysis in actual game situations is that the presence of hot hand does not necessarily have to do with what fans might have in mind when they talk about a \\player getting hot". It can

be simply the ability of specific players to hunt and exploit good matchups for them within a game, leading to a streak of successful shots" and later "exploiting missmatches..."

I don't understand what the authors are trying to express here. What this example brings to mind to me is that the short list of controls the authors use makes their results vulnerable to possible confounds, such as the type of strategic variables they are alluding to, that is not controlled for in their analysis. I´m not sure why they assume this lack of control would inflate rather than deflate estimated hot hand effect sizes. They should clarify this discussion. Bocskocsky et al, Lantis and Nesson, Miller and Sanjurjo, and perhaps Rao, have pointed out this limitation when working with field goal data.

9. The point that the authors attribute to Bocskocsky et al (2014) about players taking incrementally difficult shots on a streak of makes was made previously by Rao (2009). As mentioned above, the authors should cite Rao's important work.

10. I do not understand how controlled shooting settings, or Stephen Curry shooting at practice, or the NBA three point shootout, are not evidence of shooting performance in the "real world," as the authors claim. I believe what they mean to say is that it is in the real world but not in games. Then, game basketball settings are still basketball settings, so I think the authors should be careful not to overstate the applicability of their results, or unnecessarily make implicit critiques of controlled shooting studies. After all, controlled experiments play a pretty important role in science in general, and the trade-off between studying game and controlled shooting data is pretty obvious for people who understand a bit of statistics.

Here is another example:

"While there is literature that has examined streaks in real environments such as career trajectories and professional success [11]"

11. Should say a "large number of permutations", as in 10,000 or 25,000, rather than "a number of permutations" which is a bit too vague. Should explain clearly, somewhere, exactly how many.

7. PLOS authors have the option to publish the peer review history of their article (what does this mean?). If published, this will include your full peer review and any attached files.

Reviewer #1: No

Reviewer #2: No

---

## [Author Response · Author response to Decision Letter 1]

8 Oct 2021

(Also provided as a separate file)

We would like to thank the reviewer for their additional comments that have helped us further improve the manuscript. While we provide a point-by-point response in each comment/question from the reviewer, here we summarize the major changes we performed during this revision: 

*We have added supplementary material for describing the detailed process of performing our statistical test

*We have added supplementary material for exploring other approaches to adjusting for the shot make probability model error

*We have also expressed our conclusions more clearly and tied them with existing literature (and in particular the Lantis and Nesson paper, which somehow we had missed). As part of this point we have also made sure that we do not overstate/overgeneralize any of our conclusions/results. 

In what follows we provide a detailed response to the comments and describe the changes we have performed in the revised manuscript. 

1. The authors need to do a much better job of relating their work to the previous literature. For example, their work is quite closely related to that of Rao (2009) and Lantis and Nesson (2019). Neither of these papers is cited.

Lantis and Nesson (2019) study 12 years of NBA shooting data, including the 2 years studied by the current authors. In addition, they study both field goal shooting and free throws, as opposed to just field goal shooting. Further, they use a considerably longer list of controls than the current authors.

Their pooled test results are similar, and they don't have tests on the individual level like the current authors now do.

The authors cite Bocskocsky et al. (2014) as if it is perhaps the only regression analysis that has been performed in the hot hand fallacy literature. This is very misleading. Arkes, Lantis and Nesson, Green and Zweibel, Miller and Sanjurjo, and others use regression analysis as well.

The authors should make this clear. It is a valid, and typical, approach used in the literature. Multiple approaches have been taken to correct for the bias pointed out by Miller and Sanjurjo. The particular weaknesses of Bocskocksky et al. are not general to these studies.

An obvious question is what result would these other typical regression approaches produce with the same data. Lantis and Nesson provides an answer to this question, with more data.

We would like to thank the reviewer for pointing out these omissions in our literature review. While we were aware of Rao (2019) we felt it was more focused on the players’ perception of the existence of the hot hand - as captured from their shooting choices/behavior following a make or miss - while it was based on data from 8 players over 60 games. On the contrary, we had missed the Lantis and Nesson (2019) paper. This is a very interesting paper that overall is in agreement with our findings. In particular, they mention that when using the data from the two seasons we also use, and hence, they can control for the shot quality, overall there is a negative effect from a streak of successful shots. This is in alignment with our “league-wide” results. Furthermore, referring to Simpson’s paradox, they mention that this does not preclude individual players from exhibiting the hot hand phenomenon - something that we examined and indeed is the case for specific players. We have added a discussion for these papers in the revised manuscript. 

Furthermore, indeed we were referring to the weaknesses of the regression approach with regards to the setting at Bocskocksky et al. and not in general in the use of regression analysis in the study of the hot hand. We have clarified this in the revised manuscript. 

2. If I understood the authors' empirical approach correctly, they are comparing the computed P(M|M...M)_sim in (heterogenous Bernoulli) simulated datasets to the observed P(M|M...M)_data in the real data, rather than comparing P(M|M...M)_data to performance on the directly matched shots taken on the same streaks the real shooter was actually on. The latter approach seems more direct, and that it will be less suceptible to measurement error. The former approach may even be suceptible to systematic bias if the types of situations in which real shooters go on streaks create systematic differences in the sampling of their streak shots and those in the simulated datasets.

I suggest that the authors run the analysis I mention, and compare results with those of their current analysis, or if they agree that it is better, replace the previous analysis with it. (Later, I see new added text "We then simulate his shot sequence after k consecutive makes 100 times..."; this suggests the latter approach; this should all be explained much clearer, so there is no ambiguity)

We would like to thank the reviewer for this comment and apologize for any confusion. We are indeed comparing the simulated heterogenous Bernoulli with directly matched shots taken by the shooter in games. We have updated the corresponding text and we believe it is much more clear in the revised manuscript. 

3. Relatedly, in the revision the authors apply an ad hoc model error adjustment that samples from all shots (not just streak shot situations), then apply it to adjust hot hand estimates. First, this is a potentially serious issue with the modeling approach that the authors should address in a more complete way. Why is this happening? Is it happening systematically across shooters? If so, why? Second, if this adjustment samples at random from all shots (including shots taken on streaks) then this is creating in turn another mechanical bias against hot hand effect size. An alternative would be to sample without including these shots, which is not ideal for other reasons, but perhaps better. In either case, because this adjustment is non-trivial in size, if the authors are going to use it, they should make sure that it is not biasing hot hand effect sizes in some other more subtle way. For example, they can run simulations using different models of hot hand shooting (that is, in which they have aissumed there are hot hand shooters), and see if their model systematically produces errors in the same direction. If so, this would suggest that the measurement error they are picking up is actually a mechanical artifact of hot hand shooting, so making the correction they are could bias results against finding the hot hand.

Are the errors similar in both directions when the authors train the model in one direction, then the other?

If we are understanding this comment correctly, the main point the reviewer brings up is the fact that the model error includes any hot hand effect if present, and by using a random sample of shots to estimate the expected error for the model we could bias this estimation since it will/can include shots that have been affected by the “hot hand” (if the phenomenon exists). This essentially can lead to an underestimation of the hot hand effect. We agree with this and it is a very crucial point that we address in what follows. 

In general, the typical sources of model error are present in our shot make probability model as well, including things like omitted variables, measurement errors (e.g., in this case, shot locations might not be accurate for data obtained from specific arenas, or logging of shot types might introduce errors/inconsistencies etc.) etc. These sources of error are not systematically present in specific types of shots, but all shots in our data should be affected (or better put it is realistic we expect to be affected) in the same way. However, on top of these error sources, shots taken after a sequence of made shots might exhibit an additional error source, namely, the effect of the hot hand phenomenon (if true). Consequently, if these shots are used to estimate the expected model error - under the assumption that consecutive makes do not correlate with the success of the following shot - they can bias this estimate if the hot hand exists. 

To investigate this further we examined the model errors in more detail. We define the model error as the difference between the observed field goal percentage for a player over all his shots and his expected field goal percentage. For a given player in the set of players examined, we estimated two different types of model errors: (a) error e1 for shots following successful shots (this is the set of shots used for the hot hand analysis), and, (b) error e2 for those that do not. In both cases, the model tends to underestimate the shot make probability, with an average value for e1 = 2.2% and for e2 = 1.6%. The (paired) average difference e1 - e2 is 0.6% (the following figure provides the obtained histogram). The pooled average of these errors is approximately 1.9%, which means that on average the underestimation of the hot hand effect will be approximately 1.9-1.7 = 0.2% (if we assume that all the error term for the shots we use to examine the hot hand effect is associated with the hot hand). 

While the aforementioned shows that we might be underestimating the hot hand effect size - or even the presence of the phenomenon altogether - we are being conservative in our analysis, in the sense that we would need to have stronger evidence to reject our null research hypothesis of no hot hand. Therefore, while we keep this adjustment (i.e., using a random sample of all the shots) in the main text of the manuscript, we provide the same analysis in the supplementary material by only using shots that we do not use for the estimation of the hot hand effect. Our results agree with the aforementioned discussion, in the sense that there are now a few more players in each scenario (i.e., different value of k) exhibiting the hot hand phenomenon, but overall, when we consider the whole league, players tend to have lower FG% after a sequence of makes than what is expected from the model. 

Finally, while our results/numbers mentioned above refer to the aggregate data from both seasons in the dataset, we have examined the same for each season individually and there are not any statistical differences. 

4. in the KW example how is it possible that he only has 333 shots following at least one make, if he has taken more than 1000 shots? The only explanation that seems to make sense is that the authors are considering streaks of exactly one hit, rather than of at least one hit. But conditioning on at least one hit makes more sense because it provides more powered tests. The same goes for longer streak lengths.

We would like to apologize for any confusion. As we mention in the text, if for a specific shot there are missing features needed for the shot make probability model to make predictions or any shot for a player during a specific game is missing, then we eliminate from our dataset all the shots taken by this player in that particular game. For example, if Kemba Walker took 20 shots in a game and his shot #15 is either missing from the data or we do not have all the features to estimate a shot's probability, we remove all 20 shots from our analysis. So while we filter players with at least 1,000 shots over the two seasons that our dataset covers, the actual number of data points we can use is effectively smaller. Furthermore, assuming that there is not any systematic bias in the missing shots, players with more shots in general will also have more shots missing from the database, removing several of their games from our final dataset. In particular, for Kemba Walker we end up with a total of 844 shots. Given that his overall field goal percentage during the two seasons covered was approximately 39%, 333 shots is a consistent value for the number of shots after a make. We have updated the description of our filtering process to make this more clear. 

5. The authors should discuss how using one year to train a model to make probability estimates for another year can affect their results if there are systematic differences in shooting performance from one year to the next, e.g. the shooter improves, or changes technique over the offseason.

This is certainly one potential source of model errors (related to question 3 as well). However, we believe that in our case this does not pose a major concern for the following reasons: 

The year-to-year correlation for the field goal percentage of players is fairly high. In our data, the correlation for the players with at least 1,000 shots across the two seasons (i.e., those we used for our analysis) is 0.76. Of course, the field goal percentage is a combination of both skill and shot selection. Nevertheless, this high correlation points towards large changes in the field goal percentage from year-to-year being mainly outliers. 

While figure 1 includes the out-of-sample model evaluation for the pooled data from both seasons, we have also obtained the same evaluation results for each season individually and there is no statistical difference between the two. This is consistent with a setting where there are not significant changes in the shooting skill of player on average.

Of course, it is still possible that for specific players this setting might over or under-estimate their shot make probability. However, due to the reasons aforementioned, we do not anticipate any systematic differences on average. Anecdotally also it is extremely rare to see a player significantly improve his shooting over a single offseason (of course, a gradual improvement is highly possible -- if not expected). The only such examples that comes to our mind is (a) Lonzo Ball who shot 37.5% from the three point line during his first season with the Pelicans, after shooting 33% with the Lakers the season before, and, (b) DeAndre Jordan who made a leap in his free throw percentage during two consecutive offseasons (from 2016-17 to 2017-18 going from 48% to 58% and then in 2018-19 shooting free throws at 70.5%). We have a small discussion in the final section of the manuscript. 

6. The sampling without replacement example to explain the bias discovered by Miller and Sanjurjo is not correct. In the Econometrica they show that the bias is generally larger than the analogous sampling without replacement type effect.

We would like to thank the reviewer for this comment, which allows us to clarify the inclusion of this example. Our goal with this example is not to provide a detailed proof of the results from Miller and Sanjurjo but rather to provide the high level intuition of the streak selection bias. Now having said that, indeed, given that our example considers cases of k=1, the streak selection bias can be thought of as a form of sampling-without-replacement bias (Appendix D of the Econometrica paper), while for k > 1, there is an additional source contributing to the streak selection bias, namely, sequence overlap. The latter is absent in our example. However, as aforementioned our goal is to provide some basic intuition on the main result from Miller and Sanjurjo, which can be surprising for people unfamiliar with the area. We have clarified in the document that this example is just showcasing one source of the streak selection bias, while we provide some pointers at the Econometrica paper for more technical details for the interested reader. 

7. In response to point 3 of the other reviewer, the authors argue that an analysis of free throw shooting is not interesting for them to include because--unlike field goals---free throws are essentially identically distributed. This is not correct. Players shoot systematically better on the second ft than the first, for example.

This is indeed the case; there is a quantifiable and robust difference for the free throw percentage of the first and second free throw (about 4 percentage points) during a free throw trip. However, in our opinion, this process is still not as heterogeneous as shots taken during live ball action in a game. Most probably, if we were to order the settings with regards to heterogeneity, free throws are in-between controlled shooting experiments (more homogeneous) and live ball field goal attempts (more heterogeneous). They are an interesting setting, but we felt that adding these results could overload the paper and disrupt its flow. However, if the reviewer feels strongly about adding these results we can try and find the best way to mesh them with the current manuscript. 

8. The authors say:

"In particular, we group the shots in our test set based on their predicted probability, and for each set, we estimate the fraction of them that were actually made."

I am confused. What grouping? This should be explained clearly from the outset.

We would like to apologize for any confusion. In this part of the manuscript our objective is to evaluate the shot make probability model. In order to do that we rely on what is termed as calibration curve. Ideally, in order to evaluate a probabilistic prediction p(s) for a shot s, we would need to take that shot 100 times and count how many times the shot was made. If our probabilistic prediction was “accurate”, then we would have expected to see p(s) makes. However, taking the exact same shot - during live ball action - 100 times is not possible. This is why we take all the data points/shots in our validation/hold out set and put them into bins based on the predicted shot make probability from our model. For example, all shots for which we predicted a 55%-60% chance of being made will be placed within the same bin. Then using the data from that bin, we can estimate two quantities; (a) the (average) predicted shot make probability for all the shots in the bin, p1, and (b) the observed predicted shot make probability for the bin p2 (which is simply the FG% for the shots falling into the bin under examination). In order for the model to be well-calibrated and the probability output accurate, we need to have p1 p2. We have updated that part of the manuscript to make it more clear. 

Other Comments:

1. It is not that the streak selection bias "could lead to underestimation of the hot hand phenomenon", rather that it does lead to the underestimation.

We have changed the phrasing accordingly. 

2. The authors say:

"Inset (B) at Figure 1 presents the reliability curve for our shot probability model as obtained through an (out-of-sample) test set."

Which one? Why just one?

Given the setting we included in the first revision of the manuscript in order to expand the number of seasons analyzed, the test set includes all the shots in our dataset. In particular, the 2013-14 season forms the test set for the model trained on the 2014-15 season and vice versa. Figure 1B is presenting the results from the evaluation on these out-of-sample predictions. Possibly the easiest way to describe this is as a two-fold cross validation setting, where the folds are defined by the season. We have updated the revised manuscript to clarify this point. 

3. Need to explicitly state how many simulations of the heterogenous Bernoulli process were performed. "...repeatedly simulate" is too vague. Also the statistical testing procedure should be explained clearly.

We have made these updates in the revised manuscript, where in particular we have added a supplementary material section to describe the procedure of the statistical test as detailed as possible. 

4. The authors say:

"We use players with at least 1000 shots during the two seasons. Given that there are 82 regular season games in each season, this means that we filter out from our analysis players that took approximately less than 6 shots per game. This threshold was chosen in order to provide, on average, sequences from individual games that can be used to examine the hot-hand hypothesis for values of k > 1. As we will see in our results, we are able to examine up to k = 4."

What is this claim based on? It needs to be backed up by some evidence. Presumably the authors performed a power analysis, and they are summarizing the results here. They need to explain the results of the power analysis in a clearer way. They also suggest elsewhere that they are sufficiently powered. Based on what analysis?

We would like to thank the reviewer for this comment, which allows us to clarify our line of thought behind the choice of this threshold. We did not perform a statistical power analysis. A simple way to explain this threshold choice is that it is a necessary but not sufficient condition to perform tests for values of k up to 4. As we mention in the manuscript a threshold of 1,000 shots over two seasons leads to players with an average of a little less than 6 shots a game. If a player takes less than 6 shots in a game finding “data points” for this player with more than (k=)4 consecutive made shots - in the same game - will be very rare (indeed when we tried to look at k=5, we ended up with an average set of eligible shots for a player less than 10). So while this threshold is not a sufficient condition for having a well powered test, it is necessary for having one at all. 

5. One cannot "calculate the probability" of Kemba Walker making a shot. Should fix the language.

We would like to thank the reviewer for this comment. We have gone over the paper again and we hope we have improved/fixed the language. 

6. The authors say:

"Furthermore, in the studies by Miller and Sanjurjo on controlled shooting (both actual three-point contests [19] and a shooting field experiment [20]) they also examine whether there is hot hand activated between the different contest rounds. They achieve this by considering sequences of consecutive makes that might span different competition rounds."

This is not quite representative of Miller and Sanjurjo's approach. They first permute on the player level, to facilitate comparison with the previous literature's results, then on the player x session (or round) level, which does two things. First it eliminates vulnerability in the estimates to any systematic variation between sessions (not due to the hot hand), but if any hot hand effect at least partially activates between sessions or rounds it eliminates that too. By contrast, they are not actively trying to "examine whether there is a hot hand activated between the different contest rounds."

We would like to thank the reviewer for this detailed explanation. We have updated the part where we discuss these studies and we believe we have avoided misrepresenting their work in the revised manuscript. 

7. The authors say:

"Our results from two seasons of shooting data indicate that overall the league is subject to shooting regression, i.e., players shoot below expectation after consecutive makes, thus, regressing towards their shooting average. However, there are

players that exhibit strong statistical evidence for the presence of the hot hand individually."

This writing is not clear. First, the first statment is about the average or representative shot, across shooters. I don't understand the use of "regression towards their shooting average" here. Which shooting average?

We are simply referring here to the general phenomenon of “regression towards the mean”, according to which if a sample point is extreme, a future point is likely to be closer to the mean of the process. In our case, if a player scores, which is an “overperformance” according to the shot make probability (which is always < 1), according to the regression to the mean phenomenon his future shots are more likely to be missed, so his overall FG% performance moves closer to his personal/true average FG% (which is certainly less than the 100% observed in his last k consecutive makes). So essentially, what our results say is that if we pool together all the players, the league exhibits this reversion to the mean -- a shot taken after consecutive made shots is made at a percentage less than the expected per the shot make model. However, specific individuals show strong signals for the opposite phenomenon - which is the hot hand. We have rephrased this part in the revised manuscript to hopefully make our intended content more clear. 

8. The authors say:

"An important context that we have to add in the hot hand analysis in actual game situations is that the presence of hot hand does not necessarily have to do with what fans might have in mind when they talk about a \\player getting hot". It can

be simply the ability of specific players to hunt and exploit good matchups for them within a game, leading to a streak of successful shots" and later "exploiting missmatches..."

I don't understand what the authors are trying to express here. What this example brings to mind to me is that the short list of controls the authors use makes their results vulnerable to possible confounds, such as the type of strategic variables they are alluding to, that is not controlled for in their analysis. I´m not sure why they assume this lack of control would inflate rather than deflate estimated hot hand effect sizes. They should clarify this discussion. Bocskocsky et al, Lantis and Nesson, Miller and Sanjurjo, and perhaps Rao, have pointed out this limitation when working with field goal data.

We would like to thank the reviewer for this comment. We have updated our discussion on this in the revised manuscript but here is a more compact explanation. As mentioned in our answer to the previous comment, we simply refer to the observation that as a whole, when we pool together all the league players, a shot following (consecutive) make(s) is missed more often than expected from the shot probability model, balancing the players’ recent overperformance. What we are referring to here as a possible underlying mechanism for the players that exhibit the hot hand, stems from the purely strategic nature of the game. For example, if a team has decided to counter the opponent’s pick-and-roll with a particular coverage, this can create specific mismatches that the offense can exploit. Let’s assume (a true example from last season’s NBA playoffs) that Doncic is running the pick-and-roll and the opposing team decides to “switch”. This leads to Zubac matching up with Doncic. Even though Zubac is overall a good defender (so his model coefficient(s) will be indicative of a good defender overall) he provides a great advantage to Doncic with his ball handling and passing abilities. This type of mismatches can be exploited to a short extent - until the opposing team adjusts its strategy - and it is hard to include as controls (as we do not have all this information). Note, that while we control in the model for Zubac’s ability, this is just his overall/average ability, which can be different in specific situations. 

With regards to the controls in our model, certainly adding more controls can improve the performance of the model, but according to Lantis and Nesson (page 27 and Table 5 of the NBER working paper) adding controls beyond the shooting player’s behavior does not provide any significant improvements. Unfortunately they do not provide the detailed coefficients and standard errors for each of the independent variables considered. Similarly, Bocskocsky et al. do not provide the detailed results of their regression in order to see the relative importance of each additional covariate. In conclusion, while adding more covariates could provide predictive improvements, existing literature supports that these benefits will be minimal, so we choose to stay with the less complex model. Of course, there are variables that can significantly improve the performance of a shot make probability model (as we discuss in the manuscript) but these require information for the trajectory of the ball, a piece of information that we do not have access to. Finally, to the best of our understanding none of the existing work (Lantis and Nesson, Bocskocsky et al. etc.) have controlled/considered strategic features similar to the situation we described in the previous paragraph (which is to be expected since it is hard to identify and codify them). 

9. The point that the authors attribute to Bocskocsky et al (2014) about players taking incrementally difficult shots on a streak of makes was made previously by Rao (2009). As mentioned above, the authors should cite Rao's important work.

We would like to thank the reviewer for this comment, which we agree with. However, we did not make this point in the latest manuscript. We might have made this comment in the original version of the manuscript, which was removed after the first revision. However, according to the earlier comment from the reviewer we have added a discussion on the work by Rao (2009) in the revised manuscript.

10. I do not understand how controlled shooting settings, or Stephen Curry shooting at practice, or the NBA three point shootout, are not evidence of shooting performance in the "real world," as the authors claim. I believe what they mean to say is that it is in the real world but not in games. Then, game basketball settings are still basketball settings, so I think the authors should be careful not to overstate the applicability of their results, or unnecessarily make implicit critiques of controlled shooting studies. After all, controlled experiments play a pretty important role in science in general, and the trade-off between studying game and controlled shooting data is pretty obvious for people who understand a bit of statistics.

Here is another example:

"While there is literature that has examined streaks in real environments such as career trajectories and professional success [11]"

We certainly did not want to implicitly criticize controlled shooting studies. What the reviewer mentions is exactly what we were trying to convey; “in-game”. Controlled shooting studies are real-world experiments. We have made this clarification/distinction in the revised manuscript and used the phrase “in-game” instead of “real-world”. 

11. Should say a "large number of permutations", as in 10,000 or 25,000, rather than "a number of permutations" which is a bit too vague. Should explain clearly, somewhere, exactly how many.

We have updated this part of the manuscript -- and other locations -- to specifically mention the number of permutations/simulations performed.

---

## [Editor Report · Decision Letter 2]

14 Dec 2021

The Hot Hand in the Wild

PONE-D-21-07619R2

Dear Dr. Pelechrinis,

We’re pleased to inform you that your manuscript has been judged scientifically suitable for publication and will be formally accepted for publication once it meets all outstanding technical requirements.

Kind regards,

Pablo Brañas-Garza, PhD

Academic Editor

PLOS ONE
---

## [Editor Report · Acceptance letter]

17 Dec 2021

PONE-D-21-07619R2 

The Hot Hand in the Wild 

Dear Dr. Pelechrinis:

I'm pleased to inform you that your manuscript has been deemed suitable for publication in PLOS ONE. Congratulations! Your manuscript is now with our production department. 

Kind regards, 

on behalf of

Dr Pablo Brañas-Garza 

Academic Editor

PLOS ONE